# Anti-Arthritic and Anti-Inflammatory Effects of Andaliman Extract and Nanoandaliman in Inflammatory Arthritic Mice

**DOI:** 10.3390/foods11223576

**Published:** 2022-11-10

**Authors:** Anselmus Yakobus Lukita Adiandra Setiadi, Listya Utami Karmawan

**Affiliations:** 1Faculty of Biotechnology, Atma Jaya Catholic University of Indonesia, Jakarta 12930, Indonesia; 2Research Center for Indonesian Spices, Atma Jaya Catholic University of Indonesia, Jakarta 12930, Indonesia

**Keywords:** andaliman extract, nanoandaliman, gene expression, arthritic mice

## Abstract

Inflammatory arthritis is a severe joint disease that causes long-lasting pain that reduces a patient’s quality of life. Several commercial medicines have been used to reduce the inflammation in arthritis. However, they have side effects that affect other organs and increase the infection rate in the patient. Therefore, searching for alternative medicines from natural herbs to use as a substitute for chemical drugs and reduce the side effects of drugs has become the focus of investigation. *Zanthoxylum acanthopodium* DC., known as andaliman, is an endemic spice that originates from Tapanuli, North Sumatera (Indonesia). Our previous study confirmed that andaliman exerts anti-inflammatory and xanthin oxidase enzymatic inhibitory activities. Unfortunately, there are no in vivo studies on the efficacy of andaliman in reducing inflammation in arthritis. This research aimed to produce an andaliman extract rich in essential oils, to formulate andaliman extract in a nanoemulsion product, and to test their anti-arthritic and anti-inflammatory effects on suppressing the gene expression of inflammatory arthritis in vivo. Several steps were used to conduct this experiment, including andaliman extraction, bioactive compound identification, nanoandaliman formulation, in vivo inflammatory arthritis mice modeling using complete Freund’s adjuvant (CFA), and gene expression quantification using quantitative PCR (qPCR). Andaliman extract and nanoandaliman effectively reduced arthritic scores in CFA-induced arthritic mice. Both treatments also demonstrated anti-inflammatory potential via blocking several arthritic inflammatory gene expressions from cartilage tissue and brain in CFA-induced mice. Nanoandaliman at low dose (25 mg/kg bw) exerted a higher suppressive effect against the gene expression of *cox-2, il-ib, inos,* and *mmp-1* compared to that of andaliman extract. At high dose (100 mg/kg bw), andaliman extract effectively inhibited the expression of *il-ib, inos,* and *mmp-1* genes in arthritic mice. These data suggest that nanoandaliman may be an alternative, natural anti-arthritic and anti-inflammatory candidate for the management of inflammatory arthritis.

## 1. Introduction

Severe and chronic inflammation has become a world health problem. An untreated acute inflammation affects other organ functions, such as chronic acute inflammation due to a host immune genetic disorder, which is known as autoimmune disease and causes severe inflammation [1,2,3]. Inflammation is part of the immune system’s response to protect the body from infection by foreign substances such as virus, bacteria, cancer cells, and autoimmune disease [4]. Visually, inflammation is characterized by swelling, heat, pain, and redness in certain wounded regions [5]. At the cellular level, the cell surface recognizes antigens and activates the inflammatory pathway to release pro-inflammatory markers, which direct inflammatory and immune cells to induce inflammation in the wounded area. The body also has anti-inflammatory mechanisms to prevent the progression of acute inflammation by eliminating the antigen, reducing inflammation, and initiating the healing process [1]. Arthritis is known as a severe joint disease that causes long-lasting pain that reduces a patient’s quality of life. Several types of inflammatory arthritis are already known, such as rheumatoid arthritis, crystal-induced arthritis (gouty arthritis), septic arthritis, psoriasis arthritis, reactive arthritis, and inflammatory bowel disease arthritis [6]. 

In severe conditions, arthritis causes joint deformation, bone erosion, and dysfunction of body movement [7]. Due to the pain and severe inflammation it causes, medicines are used to reduce the pain and inflammation, and to increase the patient’s quality of life. The most well-known medicines used to medicate inflammatory arthritis are non-steroidal anti-inflammatory drugs (NSAIDs; ibuprofen) to reduce inflammation, fever and pain [8]. Meanwhile, disease-modifying anti-rheumatoid drugs (DMARD), steroids (dexamethasone), and biologic DMARDs (anti-tumor necrosis factor–α; anti-TNF-α) are used for rheumatoid arthritis. These medicines effectively reduce inflammation and the pain in the joints, but unfortunately, they have side effects that affect other organs and increase infection rates [9]. 

Herbal medicines have been widely used for reducing medicinal side effects. Indonesia is well-known for its natural resources, especially its herbs and spices [10]. Besides providing rich flavor to food, herbs and spices are also known for their use as medication or for medical treatment by their application to wounds and/or by their consumption [11]. In this study, andaliman (*Zanthoxylum acanthopodium* DC.), an endemic spice that originates from the Tapanuli region, Sumatera Utara (Indonesia) was explored for its potency for the treatment of inflammatory arthritis. This spice has a unique, lingering lemon pepper taste and local people commonly use it in their traditional cuisine, such as saksang, naniura, natiombur, arsik, and sambal andaliman [12]. It also functions as a food preservative and a medicine that has antimicrobial and anti-inflammatory activity [13,14]. Unfortunately, up until now, there are only a few in vivo studies on andaliman as an alternative natural medicine for arthritis. The latest research showed that *Z. rhetza*, a spice similar to andaliman, exerted anti-inflammatory activity for treating rheumatoid arthritis and osteoarthritis by inhibiting the gene expression of TNF-α and prostaglandin E2 (PGE2), which are related to in vitro inflammation [15]. Research on herbal medicine has focused on determining its efficacy as well as finding a better way to deliver it. Some medicine or natural substances are easily degraded due to low pH, enzymes, or vaporization, which affects their bioavailability as herbal medicines. Thus, nanoemulsions using a self-nano emulsifying drug delivery system (SNEDDS) may offer an alternative way to protect medicine from these unfavorable conditions [16]. This research aimed to test the efficacy of andaliman extract and its nanoemulsion product as an alternative medicinal candidate for protection from inflammatory arthritis in vivo, by analyzing the inflammatory arthritic gene expression using quantitative PCR (qPCR). Such information is important to determine whether andaliman extract and its nano product could become alternative natural anti-arthritic and anti-inflammatory agents to treat inflammatory arthritis.

## 2. Materials and Methods

### 2.1. Andaliman Collection and Extraction

Andaliman fruit (pericarp and seed) was purchased from the traditional market at Balige, North Tapanuli, North Sumatera (Indonesia). The fresh andaliman fruit was harvested and sun-dried by the local people. The dried andaliman were ground with a food processor and filtered using a 100-mesh filter until it was a fine powder. Andaliman extraction was done according to the method of Bahrin et al. [17] by macerating the sample with 70% ethanol (ratio sample: solvent; 1:10) in a water bath shaker at 50 °C for 3 h, and it was kept overnight in the dark at room temperature. The sample was filtered with filter paper to stop the maceration process and the 70% ethanol were separated from the sample with a rotary evaporator at 55–60 °C, 130–100 Bar, 55 rpm, until the ethanol separated and the sample was concentrated. The separated extract was dried overnight in the oven at 55 °C.

### 2.2. Formulation of Nanoemulsion Andaliman 

Nanoemulsion andaliman was formulated using the SNEDDS method [18]. The composition of the nanoandaliman product was andaliman extract, capryol-90, polyethylene glycol (PEG 400; food grade), and Tween 20. Andaliman extract from the previous step was used to formulate the nanoandaliman product. The extract was added to capryol-90, then mixed until homogenous using a homogenizer at 14,450 rpm for 10 min, then the mixture was sonicated with ultrasonic 40 kHz, and pulser power up to 60 (maximum standard usage in laboratory) for 15 min. The clear mixture was added to PEG400, then homogenized and sonicated as in the previous step until a clear mixture was produced. For the last stage, the mixture was added to Tween 20, then mixed until homogenous with a homogenizer and sonicated with ultrasonic 40 kHz and pulser power up to 60 for 10 min until it produced a clear yellow homogenized mixture.

### 2.3. In Vivo Inflammatory Arthritic Mice 

The in vivo study using inflammatory arthritis mice was done according to the modified method of Jitta et al. [19]. Male Deutchland Denken Yonken (DDY) strain mice, 7–8 weeks old was used as the arthritis inflammation model. DDY mice were given an intra-articular injection of Complete Freund’s Adjuvant (CFA) at a dose of 5 mg/mL (heat-killed and dried *Mycobacterium tuberculosis* (H 37RA, ATCC 25177) in 0.85 mL paraffin oil and 0.15 mL mannide monooleate, F5881, Sigma-Aldrich, USA) at the femoro-tibial joint of the left hind leg. The mice were observed for 2–4 weeks to confirm the inflammatory arthritis by comparing the injected CFA leg with the non-injected CFA leg from each mouse and scoring the arthritis symptoms. The arthritic score was measured by grading each leg using a 4-point scale based on erythema, swelling, and deformity of the joint to evaluate the severity of arthritis. Scores were defined as 0 = no erythema or swelling; 1 = slight erythema and swelling; 3 = moderate erythema and swelling; 4 = complete erythema and swelling, and inability to bend the ankle or wrist. 

Then, arthritis mice were treated with andaliman extract and nanoandaliman for 4 weeks. Samples were diluted in 1% of natrium carboxymethyl cellulose and administered orally at doses of at 25 and 100 m/kg. Dexamethasone (D4902, powder, CAS number of 50-02-2, purity of 98%, Sigma-Aldrich, St. Louis, MO, USA) was used as a standard drug. This drug was diluted in methanol and administered orally at a dose of 15 mg/kg. There were seven groups of mice with various treatments, each group contained 5 mice (Table 1). 

The mice were observed for 1–2 weeks after treatment and sacrificed by cervical dislocation for inflammation gene expression analysis. Before they were euthanized, blood was taken from the mice for histopathology tests and they were X-rayed to observe the arthritis inflammation. Cartilage and brain were taken. The protocol and ethics of the study were approved by the iRATco Animal Facility and IPB Animal Hospital at Bogor, West Java, Indonesia (proposal number: DR/040/17092021).

### 2.4. RNA Extraction 

Mice cartilage was collected to analyze the gene expression involved in the process of inflammatory arthritis in a mice model. A total of 100 mg of mice cartilage tissue was crushed to a fine powder with a sterilized mortar and pestle and liquid nitrogen was added before the extraction and when the sample was in the mortar. Samples were extracted with 1 mL RiboEx™ Total RNA (GeneAll Biotechnology, Korea) solution and incubated for 5 min at room temperature. Then, 300 µL of chloroform was added to the solution, homogenized with vortex for 30 s, and incubated for 4 min. Sample solution was centrifuged at 4 °C and 17,000× *g* for 15 min. After being centrifuged, it formed three phases, which were the organic phase (lower phase), debris (middle phase), and aqueous phase (upper phase). A total of 500 µL of the upper phase was moved to new vial and mixed with 500 µL isopropanol (volume ratio of RNA solution: isopropanol 1:1), followed by shaking it until RNA strands disappeared from the solution. The sample solution was then centrifuged at 4 °C and 17,000× *g* for 15 min. After centrifugation, isopropanol was discarded and 500 µL of 75% ethanol was added to the pellet, resuspended, and incubated at −20 °C for 30 min. After incubation, the pellet solution was vortexed and centrifuged again at 4 °C and 17,000× *g* for 15 min. Ethanol residue was then removed and the pellet was air-dried for 20 min. Then, 20 µL of diethylpyrocarbonate-treated water (DEPC) was added to the sample, vortexed until homogenized, and heated with a thermomixer at 55 °C for 15 min. Last, the RNA concentration was measured by Nanodrop and stored at −80 °C in a freezer until further use.

### 2.5. Quantitative PCR

The qPCR was performed by using a one-step RT-PCR Kit with Applied Biosystem^®^ 7500 based on the modified method of Adams [20]. Primers used in this study were designed using Primer3 (Table 2). A 100 ng/µL template RNA from mice cartilage was mixed with RT-PCR Master Mix (2×) Universal (KAPA SYBR^®^ FAST One-Step Universal KK4652; KAPA Biosystem, Wilmington, MA, USA), 10 µM forward primer and 10 µM reverse primer (Table 2), 50× KAPA RT MIX, 50× ROX low, and DEPC. The *gapdh,* a housekeeping gene, was used as a control while the target genes were cyclooxygenase-2 (cox-2), interleukin-1b (il-1b), inos, and matrix metalloprotease-1 (mmp-1), which are known as pro-inflammatory genes. The gene expression was analyzed using the following steps: reverse transcription, enzyme activation, denaturation, annealing or extension, data acquisition, and melting curve (Table 3). Gene expression was analyzed to identify the fold change (Fc) value of each gene target expression using ∆∆Ct, compared with K+ [21]. The ∆Ct value was used to normalize the pro-inflammatory gene target with *gapdh*, the housekeeping gene. Meanwhile, 7∆∆Ct value was applied to compare the gene expression of pro-inflammatory gene targets from each sample, and to calculate the fold change (Fc) values, to compare the change in the quantity of pro-inflammatory genes in the control treatment (K+) with treatment groups.

### 2.6. Statistical Analysis 

Data for the gene expression (Fc value) of various pro-inflammatory gene targets (*cox-2, il-1b, mmp-1,* and *inos*) from cartilage tissue and brain in CFA-induced inflammatory arthritis mice were statistically analyzed using IBM SPSS^®^ software. All experiments were performed in triplicate. The Fc value data from genes and sample treatments were arrayed by looking at the outliers. Analysis was continued with normality testing using the Kolmogrov–Smirnov test and variety testing using Levene’s test. These analysis tools were used to determine whether the data were evenly spread and uniform. If not, then the analysis was continued by using non-parametric analysis with Kruskal–Wallis, followed by a step-wise step-down post hoc test.

## 3. Results

### 3.1. Andaliman Extract and Nanoemulsion Andaliman

Andaliman ethanolic extract had a yield of 8.69% and the GC/MS profile showed that andaliman consisted of major essential oils grouped into monoterpenoid compounds, such as limonene, citronellal, geraniol, and geranyl acetate (data not shown). Nanoemulsion andaliman was formulated from andaliman ethanolic extract with the addition of capryol-90, PEG400, and Tween 20 based on the SNEDDS method. PSA data showed that this nanoandaliman had a particle size of 943.7 nm. 

### 3.2. Effect of Andaliman Extract and Nanoandaliman on the Gene Expression of Inflammatory Arthritis in CFA-Induced Arthritis Mice

The induction of CFA on arthritic mice caused an increase in the arthritis score from the first week, and sample treatment using andaliman extract and nanoandaliman in mice effectively reduced the arthritis score from the first week onward until the fourth week (Table 4). 

To assure the anti-arthritic and anti-inflammatory effects of andaliman extract and nanoandaliman, these samples were orally administered in CFA-induced arthritic mice for 4 weeks and gene expression related to inflammatory arthritis from the cartilage tissue and brain of mice were determined by qPCR. Figure 1 shows that CFA significantly increased the overexpression of *cox-2, il-1b,* and *mmp1* genes from cartilage tissue in CFA-induced mice (K+) compared to that of untreated mice (K−). Treatment with nanoandaliman (NAD) at 25 and 100 mg/kg bw resulted in a significant reduction in the expression level of *cox-2, il-1b,* and *mmp1* genes compared to that of K+ and andaliman extract (AD 25 and 100 mg/kg bw). Interestingly, dexamethasone standard (Dexa) shared a similar inhibitory pattern with NAD at 25 mg/kg bw.

Gene expression of inflammatory arthritis was also determined from brain samples in CFA-induced mice. As shown in Figure 2, CFA also induced the overexpression of *inos* and *mmp1* genes from brain in arthritic mice. Between extract and its nano product, it was found that nanoandaliman at low dose exerted a stronger inhibitory effect on *inos* and *mmp1* genes expression from brain.

## 4. Discussion

In this study, andaliman extract was formulated in a nanoemulsion and its potency as a natural arthritic therapeutic was determined. The obtained yield for andaliman extract in kinetic maceration using ethanol was 8.69%. A previous study by Suryanto et al., reported that andaliman fruit extraction using ethanol with double evaporation and a sample:solvent ratio (1:20) at room temperature produced a total yield of 7.44% [22]. Several factors may affect the extract yield, such as time, temperature, solvent type, ratio of sample and solvent, extraction method, and the sample itself [23]. The use of different parts of the plant also affects the total yield of extract. It is known that each part of plant has a different type and concentration of secondary metabolite contents due to the function of the metabolites in the plants. A study by Su et al. on various metabolites secreted from different organs of the *Acanthopanax senticosus* plant, showed that of 130 metabolites, 39 metabolites were expressed in roots, 51 metabolites were expressed in seeds, and 40 metabolites were expressed in leaves [24]. A similar study by Imphat et al. reported on the investigation of *Z. rhetsa* for anti-inflammatory effects; it showed that the extraction of *Z. rhetsa* pericarp resulted in a higher total yield of extract than the fruit and seed parts, from four different type of extraction, including hexane, ethanol (50% and 90%), decoction, and water extraction [15]. 

Other than the plant part, the usage of solvent also determines the total extraction yield. Several studies showed that ethanol extraction offered the highest total yield for Zanthoxylum plant maceration [15,22]. Essential oil is known as a non-polar compound that consists of ether, keton, aldehyde, and terpenoid. Andaliman has a unique fragrance and its essential oil has a lemon pepper aroma with a lingering after taste [25]. Non-polar solvents, such as hexane and ethanol bind easily with the essential oil of andaliman, while polar solvents such as water and acetone bind easily with the water-soluble compound from andaliman. By comparing the other study results for *Z. rhetza* extraction, it was shown that hexane gave less total yield than that of the ethanol extraction from *Z. rhetza* pericarp [15]. Our findings indicate that andaliman extract was rich in monoterpenoid compounds, such as limonene, citronellal, geranyl acetate, and geraniol, which are widely known as anti-inflammatory compounds. This result is in line with previous reports [25,26]. In addition, various plants from the Zanthoxylum genus have also been reported for their medical uses, particularly their anti-inflammatory activities, such as *Z. acanthopodium*, *Z. rhetza*, *Z. piperitum, Z. armatum,* and *Z. nitidium* [14,15,26]. Most of these plants have major monoterpenoid compounds in particular essential oils, which might be responsible for their anti-inflammatory efficacy.

To boost its potency for health promotion and protection, andaliman extract was further formulated in nanoemulsion using the SNEDDS method, and the results showed that this product had a particle size of 943.7 nm. Nanoemulsion is a water–oil–surfactant system that has multi functions to stabilize, enhance solubility, and protect the drug or plant extract as an herbal medicine from unfavorable external factors [27]. In general, mammals such as mice and humans have a low pH in the stomach and peristaltic movement in the intestine. Thus, nanoemulsion may help to increase the bioavailability of the medicine from the enzymes produced by the body [27,28,29]. Previous reports showed that *Sonchus arvensis* extract and ibuprofen were also successfully formulated and coated in nanoemulsion by SNEDDS technology [27]. 

For arthritic mice modeling, CFA was employed to induce arthritic condition, followed by scoring the arthritic symptoms. CFA was found to effectively increase the arthritic score starting from the first week of induction (Table 4). Further oral treatment with andaliman extract and nanoandaliman for 4 weeks confirmed that there was a reduction in the arthritic scores, indicating that samples were potential anti-arthritic candidates as they reduced the arthritic symptoms effectively. Among all treatments, nanoandaliman and dexamethasone significantly reduced the score compared to that of arthritic mice control. Further, the efficacy of andaliman extract and nanoandaliman were also checked at a molecular level by modulating gene expression related to inflammatory arthritis from cartilage tissue and brain in CFA-induced mice (Figure 1 and Figure 2). There were four inflammatory arthritic gene targets, including *cox-2, il-1b, inos,* and *mmp1* expressed by the targeted tissues and organs in arthritic mice. Our data demonstrated that nanoandaliman at low dose (25 mg/kg bw) showed a significant effect on suppressing the expression of *cox-2, il-1b, inos,* and *mmp1* genes, indicating its potency as an anti-arthritic and anti-inflammatory agent for the management of arthritis. 

Cyclooxygenase (COX)-2 is known as an inducible enzyme, whereas COX-2 related to inflammation [30]. If inflammation occurs, the expression of COX-2 will increase. Moreover, the increase in COX-2 will directly increase the synthesis of prostaglandin; thus, it will enhance the inflammation. Cooper et al., reported that NSAIDs as COX-2 inhibitors not only suppress the inflammation, but they also affect COX-1, which results in side effects such as stroke, gastric intestinal bleeding, hypertension, and kidney injuries [8]. In correlation with our data (Figure 1a), high expression of *cox-2* gene in arthritic mice (K+) indicated that inflammation occurred in mice cartilage treated with CFA-inducer. Lower *cox-2* gene expression in treatment groups implied that andaliman extract and nanoandaliman suppressed the inflammatory activity in mice cartilage. 

Interleukin (IL)-1b is known as a key or modulator that induces inflammatory reaction. In the normal state, no inflammation, there is only low expression of IL-1b in the body. Inflammation will increase the expression of IL-1b, which not only increases the gene expression at a molecular level, but IL-1b expression also leads macrophage and neutrophiles to the site of inflammation and maintains inflammation until the body eliminates the cause of inflammation. In some cases, the over-expression of IL-1b causes an acute inflammation that also affects other organ function [4]. In line with our data (Figure 1b), high expression of *il-1b* gene indicated that inflammation occurred in mice cartilage induced by CFA. After treatment with andaliman extract and nanoandaliman, the low level of *il-1b* gene expression implied that these samples possessed anti-arthritic and anti-inflammatory effects by suppressing the inflammatory activity in arthritic mice. 

Matrix metalloproteinase-1 (MMP-1) is a metalloproteinase enzyme that has a destructive effect on cartilage. The *mmp-1* gene expression increases when injuries occurred. It helps to destroy the matrix tissue to help inflammatory cells enter certain area to trigger an inflammation reaction and heal the wound after inflammation [31]. The *mmp-1* is also expressed as an inflammatory gene and its expression is upregulated when inflammation occurs. In this study, CFA treatment caused the high expression of mmp1 in cartilage and brain in mice, and andaliman extract and nanoandaliman as an alternative arthritis treatment significantly suppressed inflammatory arthritis symptoms in mice via reducing *mmp-1* gene expression (Figure 1c and Figure 2b). A study conducted by Tu et al., also reported that schisandrin A blocked MMP-1 and IL-1b protein expression that led to the degradation of cartilage in rat chondrocytes in vitro [32]. 

Inducible nitric oxide synthase (iNOS) is the enzyme responsible for the production of nitric oxide (NO), a major proinflammatory and destructive mediator in inflammatory diseases including arthritis [33]. High expression of iNOS is one of the direct consequences of the inflammatory process. In this study, *inos* gene expression was overexpressed in brain in CFA-induced arthritic mice, and sample treatment with andaliman extract and nanoandaliman effectively suppressed *inos* genes (Figure 2a). However, based on the quantification, this *inos* gene was not expressed by cartilage tissue of arthritic mice. It seems that the *inos* gene level was too low. A study related to non-detects in qPCR data showed that no Ct value occurred due to low signal or low gene expression of specific gene targets [33]. In terms of iNOS markers for arthritic treatment, several reports have demonstrated that common pharmacological agents, herbal, and dietary medicines have been shown to exhibit a chondroprotective effect by inhibiting the expression of iNOS direct and indirectly [34]. Therefore, the use of iNOS inhibitors for osteoarthritis treatment in human studies and clinical trials needs to be investigated.

Dexamethasone is a glucocorticoid drug for the treatment of inflammatory-related diseases, such as rheumatic arthritis, skin diseases, allergy, asthma, etc. [9]. It can be given by oral administration, muscular injection, intravenous injection, and topical administration. A previous study demonstrated that dexamethasone had a chondroprotective effect, that is, it ameliorated arthritic pain and protected cartilage from damage in a rabbit osteoarthritis model via suppressing several pro-inflammatory gene markers, including *il-1b, il-8, il-6, tnf-α, mmp-3,* and *mmp-13* [35]. In this study, dexamethasone was applied as a control drug for anti-inflammatory treatment due to its efficacy in reducing inflammation caused by arthritis mice treated with CFA. Our findings indicated that dexamethasone exerted its anti-arthritic and anti-inflammatory effects by suppressing inflammatory arthritic gene expression, including *cox-2, il-1b, inos,* and *mmp1* in arthritic mice. It is assumed that its capability was comparable with nanoandaliman at low dose (25 mg/kg). The SNEDDS system in andaliman extract ensures the product is easily absorbed and works effectively [16]. Our results demonstrated that nanoandaliman had a greater effect on reducing the level of gene expression related to inflammatory arthritis compared to that of the extract (Figure 1 and Figure 2). Due to the non-polar characteristic of andaliman extract, oral digestion makes it is more difficult for the body to absorb a non-polar compound than a nano-emulsified non-polar with oil–water emulsion [27]. Nanoemulsion helps non-polar compounds to have better solubility in terms of absorption rate and stability, which protects the extract from external factors that could affect the efficacy of the extract [28]. 

## 5. Conclusions

Andaliman extract contained several anti-inflammatory compounds from monoterpenoid groups, such as limonene, citronellal, geranyl acetate, and geraniol. A nanoandaliman product was successfully made from andaliman extract with a particle size of 943.7 nm. Our results indicate that andaliman extract and nanoandaliman show potential efficacy for use as natural arthritic therapeutics by suppressing several inflammatory arthritis gene expressions from cartilage and brain in CFA-induced arthritis mice. Nanoandaliman at low dose (25 mg/kg bw) demonstrated a greater suppressive effect against the gene expression of *cox-2, il-ib, inos*, and *mmp-1* from cartilage and brain in CFA-induced mice compared to that of andaliman extract. Therefore, nanoandaliman may be an alternative natural anti-arthritic and anti-inflammatory candidate for the management of inflammatory arthritis.

## Figures and Tables

**Figure 1 foods-11-03576-f001:**
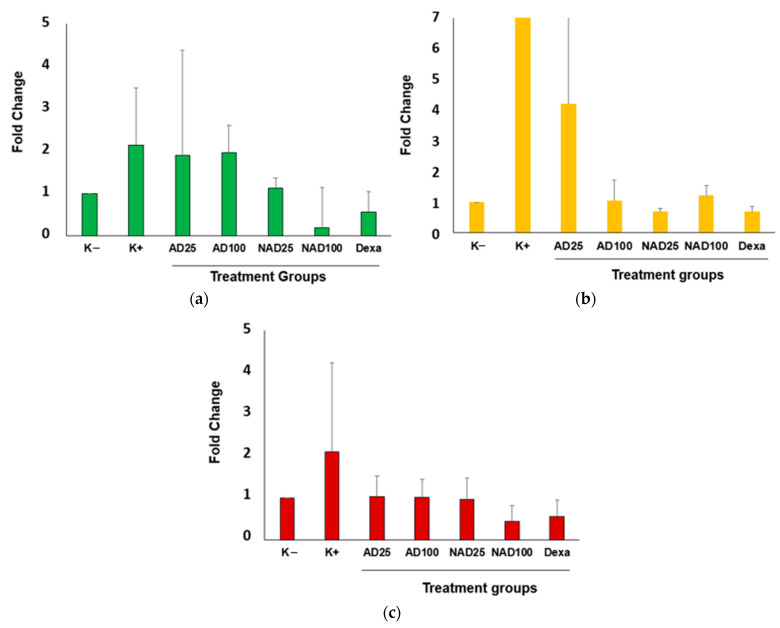
Effect of andaliman extract and nanoandaliman on gene expression of *cox-2* (**a**), *il-1b* (**b**), and *mmp1* (**c**) from cartilage tissue in CFA-induced mice. Samples were andaliman extract (25 and 100 mg/kg bw; AD25, AD100) and nanoandaliman (25 and 100 mg/kg bw; NAD25, NAD100). Dexamethasone (15 mg/kg bw; Dexa) was used as a reference anti-inflammatory drug. K−, untreated mice; K+, mice treated with CFA. P:0.233 (for *cox-2*); p:0.129 (for *il-1b* and *mmp1*) was determined by Kruskal–Wallis and continued by step-wise step-down post hoc test. There is no difference between Fc value and treatment groups. All experiments were performed in triplicate.

**Figure 2 foods-11-03576-f002:**
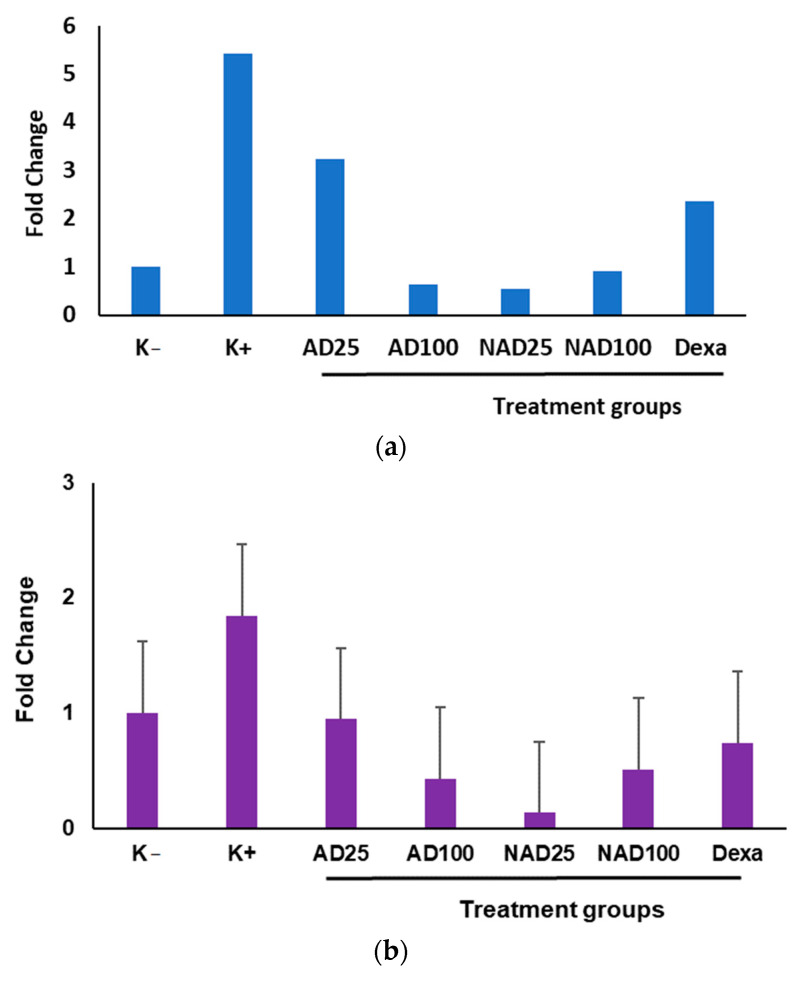
Effect of andaliman extract and nanoandaliman on gene expression of *inos* (**a**) and *mmp1* (**b**) from brain in CFA-induced mice. Samples were andaliman extract (25 and 100 mg/kg bw; AD25, AD100) and nanoandaliman (25 and 100 mg/kg bw; NAD25, NAD100). Dexamethasone (15 mg/kg bw; Dexa) was used as a reference anti-inflammatory drug. K−, untreated mice; K+, mice treated with CFA. p:0.129 (for *inos* and *mmp1*) was determined by Kruskal-Wallis and continued by step-wise step-down post hoc test. There is no difference between Fc value and treatment groups. All experiments were performed in triplicate.

**Table 1 foods-11-03576-t001:** Treatment groups of inflammatory arthritis mice.

Group	Code	Treatment
1	K−	−CFA (negative control)
2	K+	+CFA (positive control)
3	AD25	CFA + Andaliman extract 25 mg/kg
4	AD100	CFA + Andaliman extract 100 mg/kg
5	NAD25	CFA + Nanoandaliman 25 mg/kg
6	NAD100	CFA + Nanoandaliman 100 mg/kg
7	DEXA	CFA + Dexamethasone 15 mg/kg (standard)

CFA 5 mg/mL, volume of injection 50 μL.

**Table 2 foods-11-03576-t002:** Oligonucleotide primers.

Primer	Forward	Reverse
*cox-2*	5′-cttcgggagcacaacagagt-3′	5′-ggggtgccagtgatagagtg-3′
*il-1b*	5′-gagcttcaggcaggcagtat-3′	5′-tgggtgtgccgtctttcatt-3′
*inos*	5′-tgccagggtcacaactttaca-3′	5′-tgagaacagcacaaggggtt-3′
*mmp-1*	5′-gttggagcaggcaggaaggag-3′	5′-ttgcctcagcttttcagccat-3′
*gapdh*	5′-gccatcaacgaccccttcatt-3′	5′-tagactccacgacatactcagcac-3′

**Table 3 foods-11-03576-t003:** qPCR conditions.

Steps	Temperature	Duration	Cycle
Reverse Transcription	42 °C	5 min	1
Enzyme activation	95 °C	3 min	1
Denaturation	95 °C	3 s	25
Annealing/Extension	53–55 °C	30 s	
Data acquisition	72 °C	30 s	
Melting Curve	95 °C	15 s	1
	60 °C	1 min	
	95 °C	15 s	
	60 °C	15 s	

**Table 4 foods-11-03576-t004:** Arthritis score for andaliman extract and nanoandaliman in CFA-induced mice.

Groups	Arthritis Score			
	Week 1	Week 2	Week 3	Week 4
K−	0.02 ± 0.009	0.06 ± 0.004	0.06 ± 0.005	0.08 ± 0.006
K+	1.48 ± 0.160 *	1,85 ± 0.192 *	2.05 ± 0.265 *	2.44 ± 0.310 *
Dexa 15 mg/kg	1.10 ± 0.155	1.24 ± 0.250	1.12 ± 0.340	1.55 ± 0.260
AD 25 mg/kg	1.32 ± 0.156	1.57 ± 0.216	1.77 ± 0.190	1.90 ± 0.252
AD 100 mg/kg	1.25 ± 0.114	1.42 ± 0.190	1.49 ± 0.322	1.56 ± 0.140
NAD 25 mg/kg	1.21 ± 0.150	1.38 ± 0.132	1.25 ± 0.100	1.66 ± 0.128
NAD 100 mg/kg	1.02 ± 0.210	1.19 ± 0.172	1.05 ± 0.095	1.49 ± 0.112

Samples were andaliman extract (25 and 100 mg/kg bw; AD25, AD100) and nanoandaliman (25 and 100 mg/kg bw; NAD25, NAD100). Dexamethasone (15 mg/kg bw; Dexa) was used as a reference anti-inflammatory drug. K−, untreated mice; K+, mice treated with CFA. * *p* < 0.05 significant. K+ was compared with K− and the treated groups (AD, NAD, Dexa) were compared with K+.

## Data Availability

The data are avaliable from the corresponding author.

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
