# Peer review of "Anti-Arthritic and Anti-Inflammatory Effects of Andaliman Extract and Nanoandaliman in Inflammatory Arthritic Mice"

_foods, 2022, doi:10.3390/foods11223576_

Round 1

Reviewer 1 Report

The manuscript investigates the anti-inflammatory effects of an extract and a nanoformulation prepared from Zanthoxylum acanthopodium (andaliman) in an in vivo model of arthritis in mice. The study is interesting, providing some data concerning possible effects of natural compounds with anti-inflammatory properties. However, in its current form, the manuscript requires extensive modifications. Major points of concern:

1. The manuscript does not provide a proper phytochemical analysis of the extract. If a GC/MS analysis was performed, some quantitative data should be presented concerning major terpenoid compounds.

2. The methods for animal study are not adequately presented. How was the score for grading arthritis been established (no. of points for each affected joint)? Where was the injection site of CFA, tibiotarsal joint or elsewhere?

3. In the discussion section of the manuscript the effects on gene expression of key inflammatory mediators are not properly linked with individual chemical constituents, although a variety of studies were published before. Moreover the authors claim that  "schisandrin A blocked MMP-1 and IL-1b protein expression" but is this compound present in their extract??

4. There are major pharmacologic mistakes in the manuscript. An example from Line 324 "Dexamethasone is known as a steroid disease modifying anti-rheumatoid drugs 324 (DMARD)". The authors should know that corticosteroids are not DMARD drugs. Please read again reference no. 9.

5. The manuscript is full of English language errors, mistakes and bizzare choice of words which should be corrected (animals were "executed" etc). 

Author Response

Jakarta, 19 October 2022

To:

Reviewer 1

Dear Sir,

Enclosed please find our revised manuscript entitled: “Anti-arthritic and Anti-inflammatory Effects of Andaliman Extract and Nanoandaliman in Inflammatory Arthritic Mice”. Based on the reviewer’s suggestion and valuable inputs, we have revised and responded to each comment carefully in the attached PDF file. Please also find the revised version of manuscript.

Sincerely,

Yanti

Reviewer 2 Report

1) Mention the details of complete Freund’s adjuvant (CFA) and Dexamethasone (such as purity, CAS no., Stock solution preparation) in the materials, and method

2) Mention the details of the RNA extraction kit, and rtPCR kit (such as CAS no., and the name of the company).

3) Give the proper justification for using an RNA template for rt PCR, Since cDNA synthesized from RNA was found suitable for rtPCR.

4) Describe the GC/MS run method, and sample preparation of Andaliman ethanolic extract and Show the GCMS profile.

5) In line no.175, the author mentioned a yield of 8.69%. Describe the name of the individual or mixture of compounds in the 8.69% yield.

6) Mention the biological repetition in the legend of Fig. 1 & 2

7) Give justification for why nanoemulsion of dexamethasone is not included in the treatment. 

8)  The NSAIDs used to treat inflammation are not selective and will block both cox 1 & 2. The treatment of Nanoandaliman at 25 mg/kg significantly reduces the expression of all target genes. Authors need to describe the effect of this treatment on the expression of COX-1 gene also.   

Author Response

To:

Reviewer 2

Dear Sir,

Enclosed please find our revised manuscript entitled: “Anti-arthritic and Anti-inflammatory Effects of Andaliman Extract and Nanoandaliman in Inflammatory Arthritic Mice”. Based on the reviewer’s suggestion and valuable inputs, we have revised and responded to each comment carefully in the attached PDF file. Please also find the revised version of manuscript.

Sincerely,

Yanti

Round 2

Reviewer 1 Report

The authors have revised the manuscript according to suggestions, therefore it can be published in the current form.